# Numerical investigation of the plastic deformation behaviour of graded crushed stone

**Ying-jun Jiang, Chen-yang Ni** [ID]**\*, Yu Zhang, Yong Yi**

Key Laboratory for Special Area Highway Engineering of Ministry of Education, Chang'an University, Xi'an, China

\* chenyangni@chd.edu.cn

## Abstract

A numerical model of the dynamic triaxial test of graded crushed stone was established based on DEM (Discrete Element Method) to study its dynamic characteristics. The influence of test conditions on simulation results was analysed through numerical simulation. A method for determining the test conditions was proposed, and the reliability of the simulation was verified. We studied the accumulation rule and failure standard of permanent deformation. We then determined the critical failure stress level. The prediction model of permanent deformation cumulative failure was then established. When the calculation time step is greater than $1E-4$ s per step, the stability of the dynamic triaxial numerical test of graded crushed stone is good. The plastic deformation of the simulated specimen tends to be stable under 10,000 dynamic loading cycles. When the specimen height and diameter are greater than 20 and 10 cm, respectively, the specimen size has little influence on the simulated axial strain. The recommended specimen size is 10 cm (Φ) × 20 cm (h). The action time curve results are consistent with indoor measurement results, which proves the simulation's reliability. The critical failure stress is approximately linearly correlated with the confining pressure, and the cumulative failure equation of plastic deformation is established.

## 1 Introduction

Graded crushed stone is often used in road pavement bases. It is immensely significant to study the dynamic and deformation characteristics of graded crushed stone to improve pavement durability [1]. Janoo and Bayer experimentally analysed the properties of graded crushed stone raw materials, divided the evaluation index and grade of raw materials and analysed the influence of different factors on the dynamic modulus and shear strength of graded crushed stone through triaxial tests [2]. Brown and Chan studied the nonlinear characteristics of graded crushed stone using many laboratory tests and fitted the recommended regression equation and parameter range through a large amount of data [3]. Rada and Witczak studied the behavior law of plastic deformation and rebound deformation of graded gravel through an indoor dynamic triaxial test [4]. Azam et al. hypothesized that the final plastic deformation would reach a limit value and that the permanent deformation of graded crushed stone would

**Funding:** This work was supported by the Scientific Research of Central Colleges of China for Chang'an

University (No. 300102218212); the scientific project from Shaanxi Provincial Communication No. 19-27K and the scientific project from Shaanxi Provincial Communication No. 18-02K.

**Competing interests:** I have read the journal's policy and the authors of this manuscript have the following competing interests.

accumulate to the limit value when the number of loading actions tends to be infinite [5]. Werkmeister et al. established a calculation model of graded crushed stone pavement using finite element software and combined it with an indoor dynamic triaxial test to calculate the permanent deformation. The orthogonal method was used to analyse the change law of water content, gradation, raw material characteristics, load and stress level [6]. Wang et al. studied the development law and distribution state of plastic deformation of graded crushed stone base under long-term vehicle load [7, 8]. Wang et al. analysed the dynamic deformation characteristics of graded gravel mixtures [9]. Yang et al. explored the influence of fine particle content on the dynamic deformation behaviour and cumulative plastic strain of graded gravel [10].

Because of the obvious nonlinear mechanical properties and discontinuous medium structure of graded crushed stone, the relationship between mechanical parameters, experimental conditions and deformation performance of graded crushed stone cannot be theoretically analysed in detail owing to the limitation of current mechanical theory. In addition, the traditional indoor test method has low efficiency and high cost, so the research depth and breadth of the traditional indoor test method for studying the dynamic characteristics of graded gravel are limited. To solve the abovementioned problems, the DEM is widely used for studying mechanical properties of road materials, which provides a theoretical basis and method support for the study of dynamic characteristics of graded crushed stone [11–15]. Therefore, based on DEM and particle flow theory, the dynamic triaxial numerical test method of graded crushed stone is constructed in this work to study the plastic deformation law and critical failure stress of graded gravel, and the cumulative failure equation of plastic deformation of graded gravel is proposed.

## 2 Materials and testing method

### 2.1 Materials

The raw material used in this work was Yinghu limestone from Ankang, Shaanxi Province. The apparent density of the raw material is given in Table 1; the mineral aggregate gradation is given in Table 2. A, B and C are the median values of skeleton dense gradation, continuous gradation 1 and continuous gradation 2, respectively, in JTG D50-2015 code for the design of highway asphalt pavement (hereinafter 'the specification').

### 2.2 Testing method

**2.2.1 Test conditions.** *(1) Test instrument.* In this paper, UTM-25 (universal testing machine) was used to conduct the indoor dynamic triaxial test, as shown in Fig 1.

*(2) Loading time and loading waveform.* Loading time and loading waveform are important parameters for dynamic testing of permanent deformation of loose aggregates. The load waveform and loading time of the dynamic test should be determined according to the response law of the pavement under the action of the traffic load. When the load is quite far away from a certain point on the road, the point's stress is 0, and when the load directly acts on the point, the stress reaches the maximum. It is assumed that with change in time, the load strength

**Table 1. Apparent density of Yinghu limestone from Ankang.**

| Aggregate size (mm) | 19.0–31.5 | 9.5–19.0 | 4.75–9.5 | ≤4.75 |
|---|---|---|---|---|
| Apparent density (kg·m$^{-3}$) | 2,712 | 2,709 | 2,692 | 2,681 |

**Table 2. Mineral aggregate gradation.**

| Gradation | Percentage by mass passing through the following sieves (mm) (%) | | | | | | |
|---|---|---|---|---|---|---|---|
| | **31.5** | **19.0** | **9.5** | **4.75** | **2.36** | **0.6** | **0.075** |
| A | 100 | 73 | 36 | 36 | 23 | 11 | 3 |
| B | 100 | 73 | 53 | 38 | 29 | 17 | 4 |
| C | 100 | 85 | 59 | 43 | 29 | 17 | 4 |

exhibits the law of half-sine function; the expression for load strength is given by formula (1):

$$L_t = qsin^2\left(\frac{\pi}{2} + \frac{\pi t}{d}\right),$$ (1)

where $d$ is the load action time (s) and $q$ is the load strength (MPa).

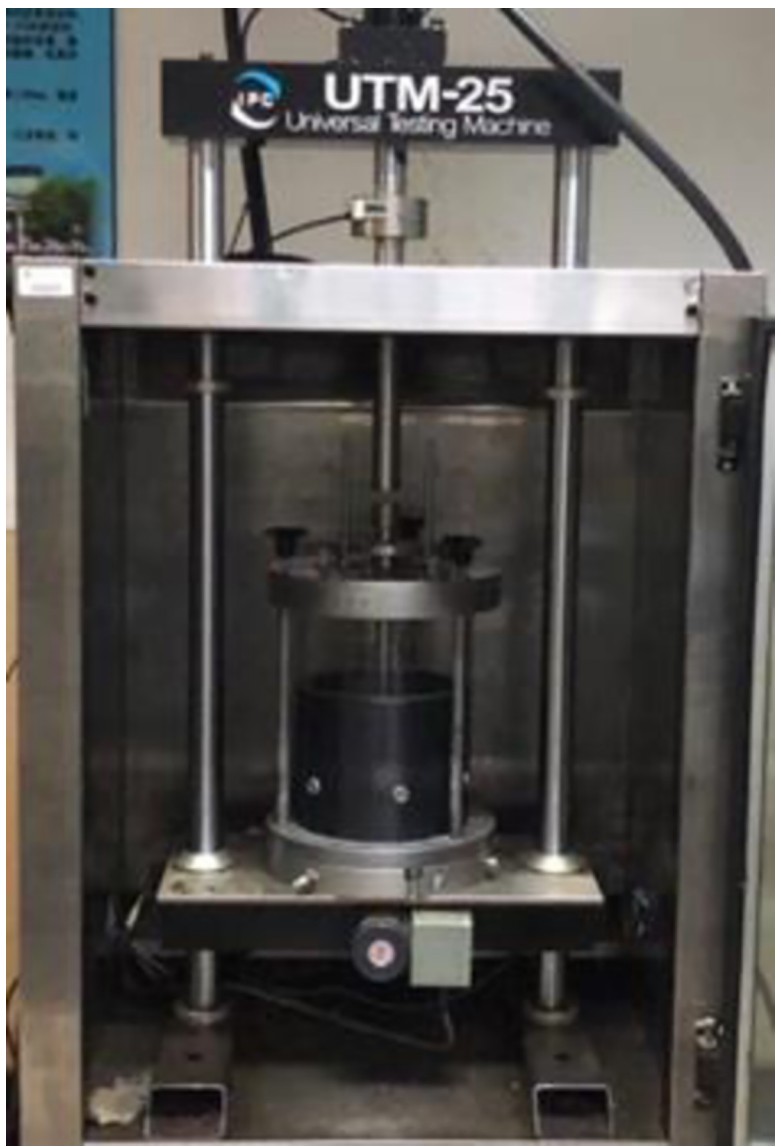

**Fig 1. Indoor dynamic triaxial test using UTM-25.**

When the load is far away or $L_t = \pm\frac{d}{2}$, then $L_t = 0$; when the load directly acts on $t = 0$, the load strength reaches the maximum value.

The time of load action depends on vehicle speed v and tyre contact area radius a. Previous studies have shown that when the load distance is six times that of the tyre grounding area radius, it can be considered that the load does not affect the road surface. The specific expression is given by formula (2):

$$d = \frac{12a}{v},\qquad(2)$$

where $v$ is the vehicle speed (m s−1) and $a$ is the tyre contact area radius (m).

The smaller the vehicle speed $v$ is, the longer the load acts on the road surface, and the greater the impact is. The known v is 64 km·h$^{-1}$, and the load action time is 0.1 s. Therefore, considering the most adverse effect, this paper selects the half-sine wave with dynamic load action time $t_z = 0.1$ s and dynamic load action time interval $t_j = 0.9$ s and cycle 1.0 s. The load waveform curve is shown in Fig 2.

*(3) Confining pressure.* Before the dynamic triaxial test, ANSYS was used to calculate the stress distribution of each structural layer of an asphalt pavement when the normal tyre ground pressure is 0.7–1.1 MPa within the elastic–plastic theory range. The horizontal stress distribution in the granular layer is generally between 20 and 150 kPa. To simulate the actual confining pressure of graded crushed stone, the maximum σ3 is slightly greater than the maximum effective stress actually borne by the structural layer. The minimum σ3 should not be less than the pressure experienced by the structural layer. The confining pressure ranges from 50 to 150 kPa (50, 75, 100, 125 and 150 kPa).

*(4) Deviator stress.* The ultimate strength in static triaxial tests under different confining pressures was divided into five stress levels to compare the plastic deformation characteristics of crushed stones under long-term dynamic loading. For the elastic–plastic loose aggregate of graded crushed stone, on the one hand, its stress–strain curve exhibits nonlinear change: the greater the load, the more prominent the plastic performance of mixture. On the other hand, the loading stress level selection should consider the stress state of a graded gravel layer asphalt pavement structure under general vehicle loads. Therefore, for the repeated loading test of permanent deformation of graded crushed stone, the stress level was selected between 0.1 and 0.9, and the stress level S = 0.1, 0.3, 0.5, 0.7 and 0.9 was selected in this work.

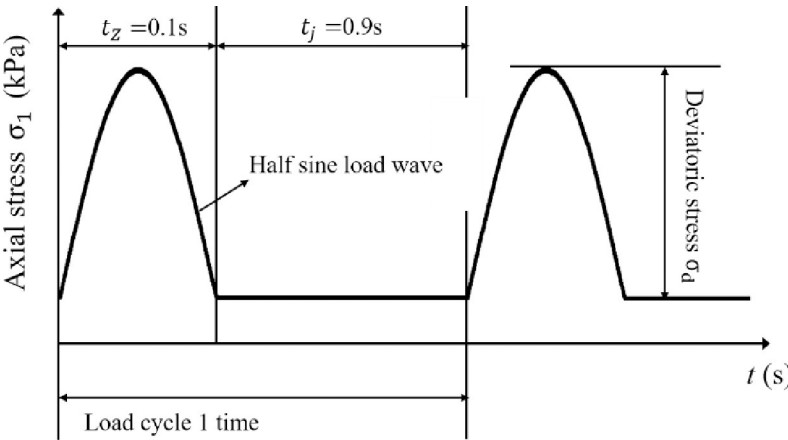

**Fig 2. Loading waveform.**

S is the ratio of maximum cyclic stress to shear strength, as given by formula (3):

$$S = \frac{\sigma_{max}}{\sigma_s},$$

(3)

where $\sigma_{max}$ is the maximum cyclic stress (kPa) and $\sigma_s$ is the ultimate strength of the static triaxial test (kPa).

*(5) Number of loading actions.* The number of loading actions is an important factor for studying the permanent deformation behaviour of graded crushed stone. Based on existing research, the number of loadings in the repeated loading test in this paper is 10,000 [16–24].

**2.2.2 Test procedure.** This test refers to the AASHTO DESIGNATION T307-99 test method, and the specific steps are as follows:

1. After the specimen is prepared, place it on the test bench with both ends of the rubber membrane tightly tied to the base; install the pressure chamber cover, connect the loading shaft and the actuator and inject silicone oil until the entire pressure chamber is filled.

2. Install the axial differential displacement metre and connect the piezoelectric pressure sensor.

3. Set the loading scheme and set the confining pressure until the required value is stable for 30 min.

4. When the deformation of the specimen is stable under static axial stress, the dynamic loading module is used to apply the dynamic load, and the stress control method is used; that is, the deviator stress remains unchanged during the test. When the specimen is loaded 10,000 times, the deformation stops. Remove the confining pressure, remove the test piece and prepare for the next round of tests.

5. Change the next specimen, change the deviator stress and confining pressure, repeat steps (1)–(4) and sort the data.

# 3 Dynamic triaxial numerical test of graded crushed stone

## 3.1 Construction of the numerical test method

**3.1.1 Generation of graded crushed stone.** The DEM built-in command 'wall' was used to generate four 'walls' to form a closed rectangular area for simulating the test mould. Assuming there are *n* types of crushed stones in the range of particle size to form the mixture according to a certain gradation, the surface mass *m* of the specimen in a two-dimensional plane state equivalent to *M* is as follows:

$$m = \frac{M}{V} \times S = \frac{V\rho_{max}}{V} \times D \times h = Dh\rho_{max},$$

(4)

where $M$ is the total mass of the test piece (g), $M = \rho_{max} \times V = \rho_{max} \times \frac{\pi D^2}{4} \times h$, $\rho_{max}$ is the maximum dry density (g·cm$^{-3}$), D is the diameter of the test piece (cm) and h is the height of the test piece (cm), $m$ is the surface quality of a specimen (g) and $S$ is the two-dimensional mapping area of the specimen (cm$^2$).

We calculate the two-dimensional mapping area $S_i$ of the type $i$ gravel as

$$S_i = \frac{m_i}{\rho_i} = \frac{DhKP_i}{\rho_i}\rho_{max},$$

(5)

where $S_i$ is the two-dimensional mapping area of type $i$ gravel (cm$^2$) and $\rho_i$ is the density of

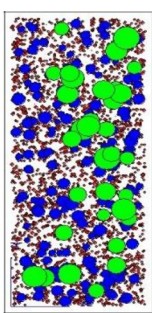
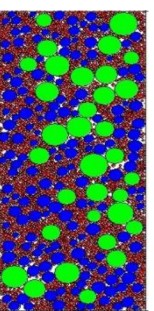

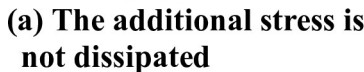

**(a) The additional stress is
not dissipated**

**(b) The dissipation of
additional stress is complete**

**Fig 3. Aggregate model.** (a) The additional stress is not dissipated. (b) The dissipation of additional stress is complete.

type $i$ crushed stones (g·cm$^{-3}$), $m_i$ is the quality of the gravel surface of type $i$ (g), $m_i = m \times P_i \times K = DKhP_i\rho_{max}$, $P_i$ is the percentage of the sieve residue of the $i$-type crushed stones (%) and $K$ is the compactness (%).

Within the scope of the test mould, various specifications of crushed stones were generated to form a mixture, as shown in Fig 3.

It can be seen from Fig 3 that because the centre coordinates of aggregate particles are generated randomly, particle overlapping will inevitably occur in the finite space of the simulated test mould, and significant additional stress will be generated between particles, as shown in Fig 3(A). To eliminate this adverse effect, the generated 'wall' should keep a certain distance from the particles so that they have enough space to dissipate their own additional stress at the beginning of the calculation cycle. See Fig 3(B) for the moulded specimen after additional stress dissipation.

**3.1.2 Formation of the indenter and the diaphragm.** A row of spheres is generated on both sides of the 'wall' on the top and bottom surfaces. The sphere and sphere are connected as a whole by bond model. The dynamic triaxial test indenter of graded gravel is simulated. The original two 'wall' are given acceleration to the horizontal two sides, and the two 'wall' are pulled out to both sides along the horizontal direction. We kept the left and right 'walls' still and simulated the dynamic triaxial test rubber membrane of graded gravel to simulate the application of confining pressure, as shown in Fig 4.

## 3.2 Sensitivity and determination of test conditions

To determine the standard test conditions for the dynamic triaxial numerical test of graded crushed stone, the sensitivity of the results of the dynamic triaxial numerical test of graded crushed stone to the calculation time step, loading times and specimen size was investigated in this work. See Fig 5 for the loading diagram of the dynamic triaxial numerical test of graded crushed stone.

**3.2.1 Influence of calculation time step.** The calculation time step reflects the relationship between computer simulation time and the actual time. In principle, the calculation time step should not exceed the default value of the computer. However, the larger the calculation time step, the more information contained in the same time step and the greater the amount of calculation. It is proven that the selection of the calculation time step has a great influence on the stability of the numerical test [25]. Different calculation time steps $dt$ were selected to stabilise the virtual test piece, and the influence of the calculation time step on the stability of the numerical test was analysed to select a reasonable calculation time step. See Fig 6 for the simulation diagram.

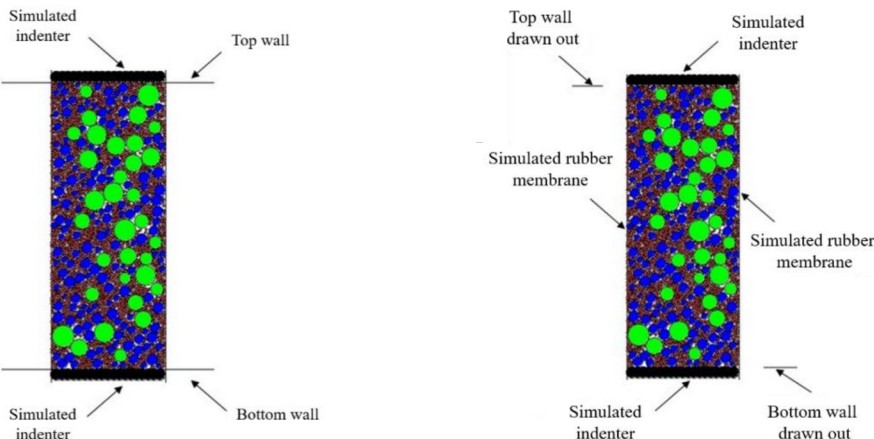

**(a) Test piece before 'wall' extraction**　　**(b) Test piece after 'wall' pulling out**

**Fig 4. Virtual specimen for the dynamic triaxial numerical test of graded crushed stone.** (a) Test piece before 'wall' extraction. (b) Test piece after 'wall' pulling out.

It can be seen from Fig 6 that when the calculation time step is greater than $1E−4$ s per step, the numerical test effect is stable and the simulated indenter has no major disturbance. When the calculation time step is $1E−3$ s per step, the upper simulated indenter flies away from the virtual specimen range. In the specimen range, the lower simulated indenter tilts, and a small number of aggregates fly out to both sides of the specimen. When the calculation time step is $1E−2$ s per step, the upper and lower indenters fly away from the virtual specimen range, and many aggregate flies out to both sides of the specimen. To ensure the stability of the numerical experiment while considering the computational efficiency of the numerical experiment, the numerical experiment calculation step was taken as $1E−4$ s per step.

**3.2.2 Influence of load times.** The dynamic triaxial simulation test method was used to study the influence of the number of loading actions on the permanent deformation of the graded crushed stone. The microcontact force state of the virtual sample of the graded crushed stone upon application of a dynamic load is shown in Fig 7. The number of loading actions is 100,000. The relationship between the permanent strain $\varepsilon$ of the graded crushed stone and the number of loading actions $N$ is shown in Fig 8.

Fig 7 shows that when the virtual sample of graded crushed stone is not subjected to dynamic loading, the direction of load transmission is not obvious. The distribution of the contact force network is disorderly in the entire specimen. When the virtual sample of graded gravel is subjected to dynamic load, the contact force concentration phenomenon appears at the top and bottom of the specimen. By contrast, the distribution of contact force networks in the virtual specimen does not change. With increase in the number of loadings, we found that the contact force is transmitted along the main skeleton in a strip form. When the virtual specimen of graded crushed stone is in the dynamic load interval stage, the contact stress concentration inside the specimen increases and the stress concentration at the top and bottom decreases. It can be seen that the dynamic load not only acts on the surface of the virtual specimen but also has a significant effect on the interior of the specimen.

It can be seen from Fig 8 that the axial permanent strain of graded crushed stone materials increases significantly with the increase in loading times (especially the first 1000 times). When the loading times reach 10,000, the axial permanent strain growth is no longer obvious and gradually tends to be stable. The continuous increase in loading times has little effect on

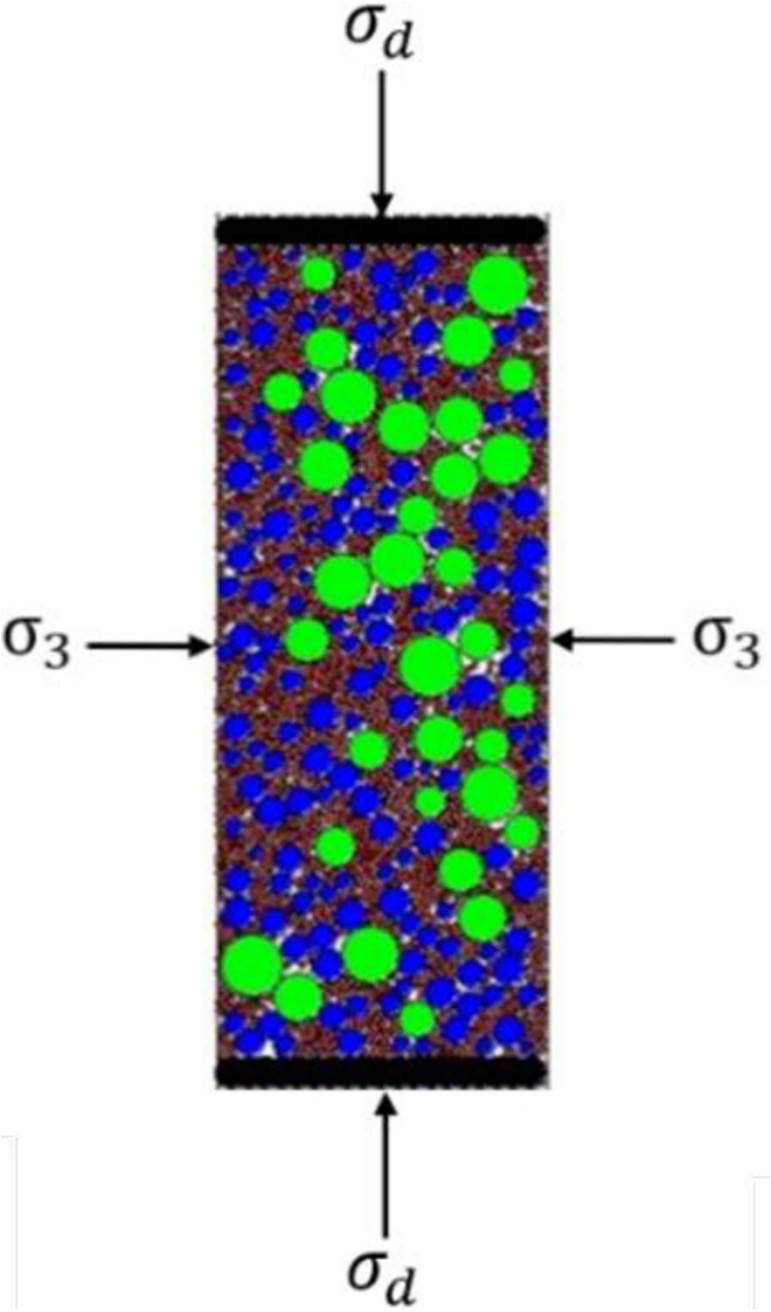

**Fig 5. Loading diagram of dynamic triaxial numerical test for graded crushed stone.**

the axial permanent strain. Therefore, to simplify the test and comprehensively monitor the influence of loading times on axial permanent strain, the dynamic load action times of 10,000 was adopted in the indoor permanent deformation test of graded crushed stone.

**3.2.3 Influence of specimen size.** *(1) Height of the test piece.* Fig 9 shows the change rule of $\varepsilon_{10000}$ of graded crushed stone with respect to the height of the test piece ($h$). The stress condition of the test is that the confining pressure is 50 kPa, the diameter of the specimen is $\Phi = 10$ cm, and the nominal maximum particle size of the aggregate is abbreviated as $D_{\max}$.

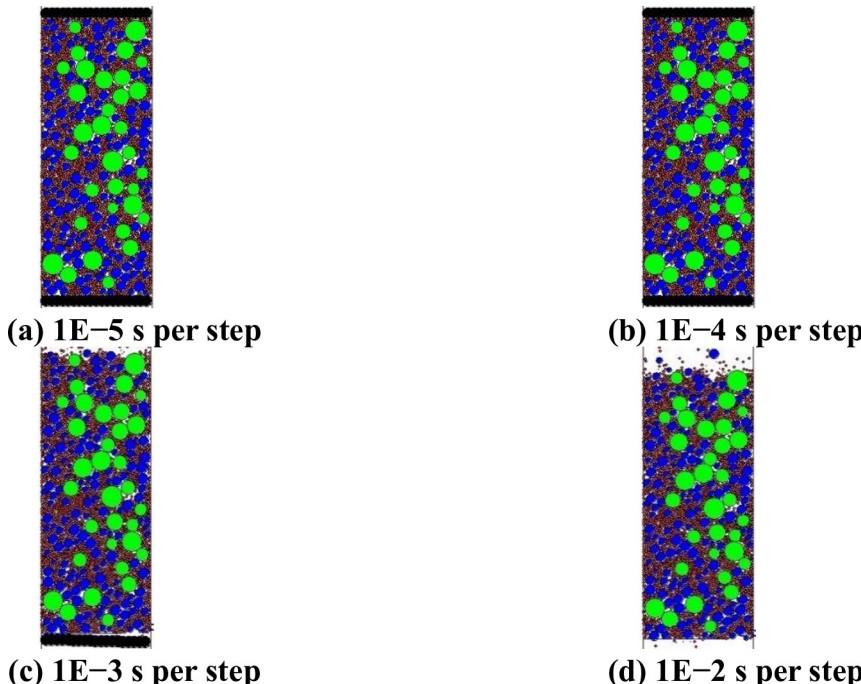

**Fig 6. Diagram of voltage stabilisation effect at different calculation time steps.** (a) 1E−5 s per step. (b) 1E−4 s per step. (c) 1E−3 s per step. (d) 1E−2 s per step.

It can be seen from Fig 9 that the influence of the height ($h$) of the graded crushed stone specimens with different maximum nominal sizes $D_{max}$ on $\varepsilon_{10000}$ is similar. With increase in specimen height, $\varepsilon_{10000}$ decreases first and then tends to be stable. When $D_{max} \leq 31.5$ mm, when $\varepsilon_{10000}$ tends to be stable, H $\geq$ 20 cm; when $D_{max} = 37.5$ mm, when $\varepsilon_{10000}$ tends to be stable, $h$ = 15 cm. Therefore, when $h \geq$ 20 cm, the gravel particle size tends to be stable, and the size of the specimen has little effect on the numerical test results.

For further analysis, this paper presents the appropriate displacement field of the virtual specimen at different specimen heights, as shown in Fig 10.

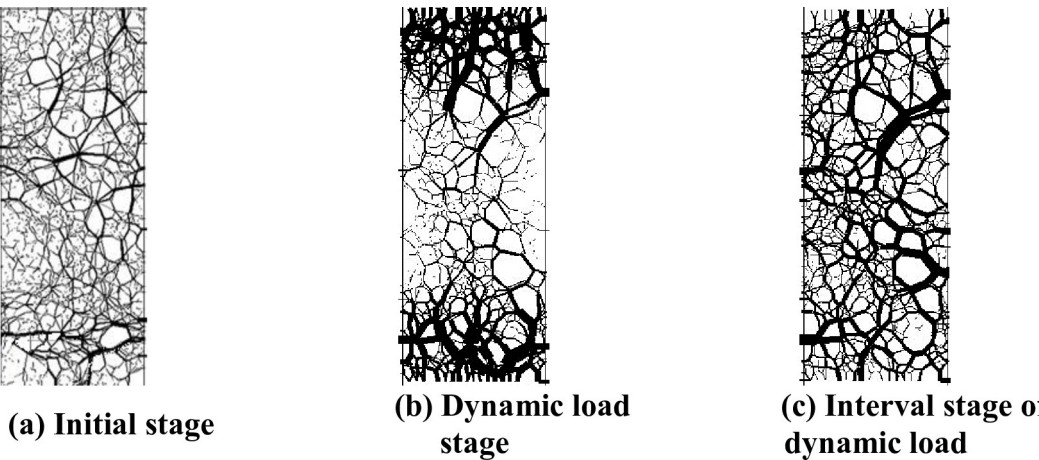

**Fig 7. Contact force state of graded crushed stone in dynamic triaxial simulation test.** (a) Initial stage. (b) Dynamic load stage. (c) Interval stage of dynamic load.

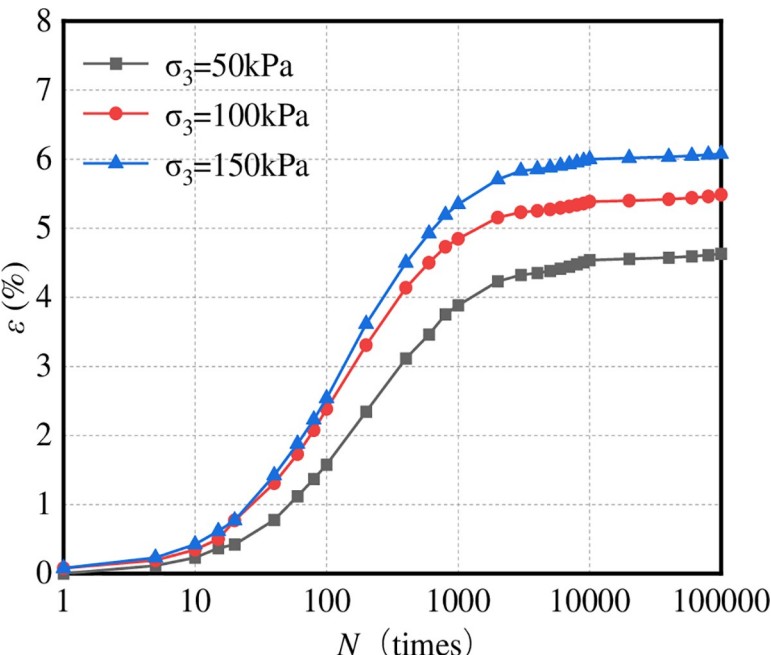

**Fig 8. ε–N.**

It can be seen from Fig 10 that when $h \leq 15$ cm, the aggregate in the virtual specimen has obvious displacement, and the aggregate in the middle of the test mould moves violently and accumulates towards the middle of the specimen. This is because the upper and lower parts of the aggregate are disturbed when the indenter acts on the aggregate, resulting in opposite vertical displacement. When 20 cm $< h <$ 25 cm, because of the increase in specimen height, the

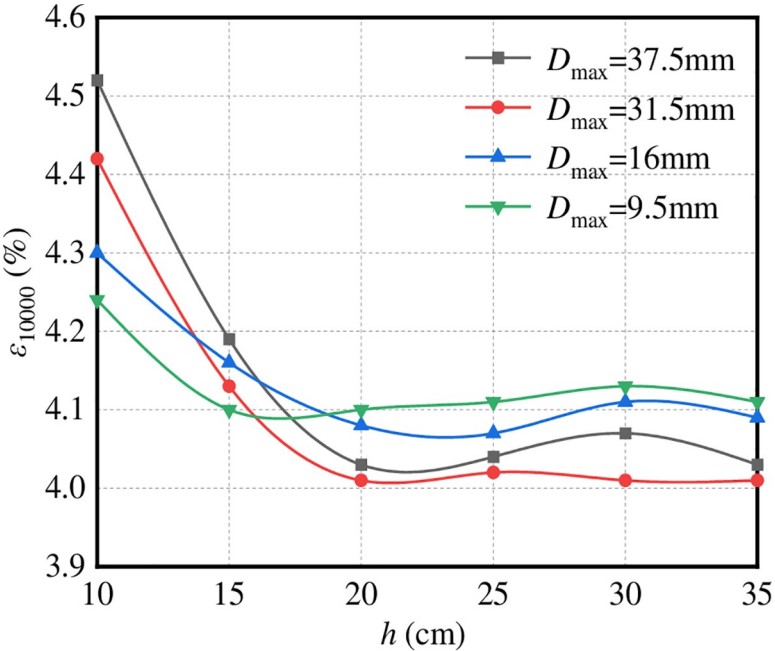

**Fig 9. $\varepsilon_{10000}$–$h$.**

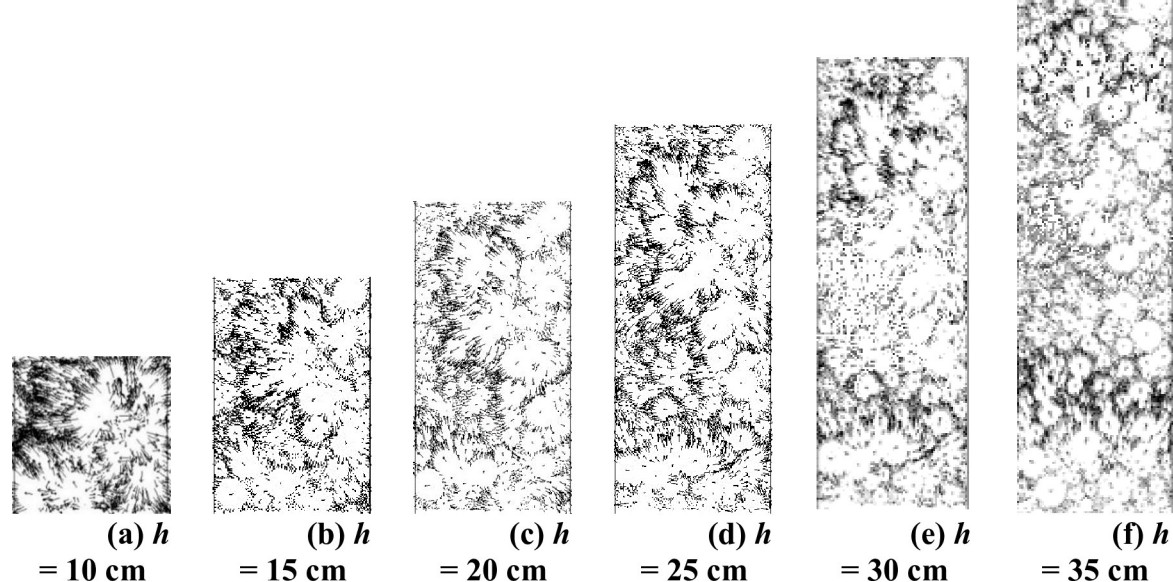

**Fig 10. Meso displacement field of virtual specimens with different heights.** (a) $h$ = 10 cm. (b) $h$ = 15 cm. (c) $h$ = 20 cm. (d) $h$ = 25 cm. (e) $h$ = 30 cm. (f) $h$ = 35 cm.

displacement vector has been partially consumed, which results in the aggregate in the middle of the specimen moving laterally but not to the middle. When $h \geq 25$ cm, a 'weak influence area' appears in the middle area of the test mould, and the aggregate displacement is minimal in this area. This is due to the continuous increase in the specimen height and because the displacement vector has been exhausted before the opposite moving aggregates meet, which prevents the indenter from disturbing the aggregate at this time. The axial strain only comes from the compaction of the upper and lower parts of the aggregate, so the axial strain is small.

To sum up, when the height of the specimen is small, the transverse displacement of the aggregate and the axial strain are too large, which obviously cannot give full play to the performance of the graded crushed stone material. When the height of the test piece is too large, there is a 'weak influence area' in the middle of the test mould. Therefore, with respect to the height of the test piece, it is necessary to ensure that the axial strain is within the reasonable range and avoid the occurrence of a 'weak influence area'. It is suggested that the specimen height $h$ should be 20–25 cm.

*(2) Diameter of the test piece*. Fig 11 shows the change rule of $\varepsilon 10000$ of graded crushed stone with respect to the diameter of the test piece ($\Phi$). The stress condition of the test is that the confining pressure is 50 kPa and the height of the specimen is $h$ = 20 cm.

It can be seen from Fig 11 that the influence of the diameter of graded crushed stone specimens with different nominal maximum particle sizes $D_{max}$ on $\varepsilon_{10000}$ is similar. With the increase in specimen diameter, $\varepsilon_{10000}$ first decreases and then tends to be stable. When the maximum particle size $D_{max} \geq 9.5$ mm, when $\varepsilon_{10000}$ tends to be stable, $\Phi \geq 15$ cm; when the maximum particle size $D_{max} \leq 9.5$ mm, when $\varepsilon_{10000}$ tends to be stable, $\Phi$ = 10 cm. At this time, when the particle size is larger than or equal to 15 cm, the test results are slightly affected by the particle size.

For further analysis, the microdisplacement vector field of the virtual specimen with different specimen diameters is shown in Fig 12.

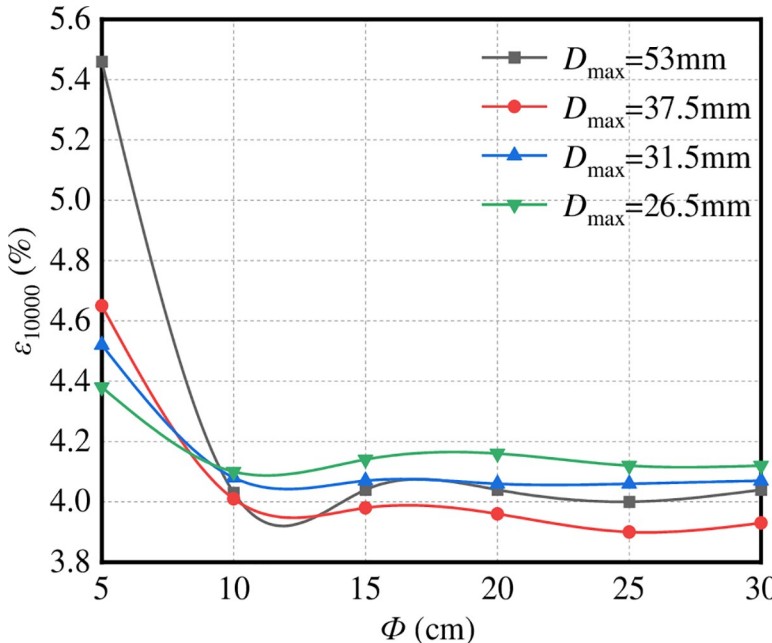

**Fig 11. $\varepsilon_{10000}$–$\Phi$.**

It can be seen from Fig 12 that the aggregate particles in specimens with different diameters have different degrees of lateral movement. When the diameter is less than or equal to 15 cm, the upper part of the aggregate particles in the specimen clearly exhibits a 45˚ oblique downward shear plane. Because of the constraints of the test moulds on both sides of the specimen, the vertical displacement of the particles on both sides of the specimen is large. When the specimen diameter is greater than or equal to 20 cm, the transverse movement of the aggregate particles mainly occurs in the middle of the specimen, and the displacement of the aggregate particles on both sides of the specimen is minimal, which is due to the action of the indenter. However, because of the large diameter of the specimen, the displacement vector is largely dissipated before the displacement vector is transferred to the two sides of the test mould. The displacement of the aggregate on both sides of the specimen is minimal, and the 'weak influence area' appears on both sides of the specimen, so the axial strain is small.

To sum up, when $\Phi \geq 20$ cm, there is a 'weak influence area' on both sides of the specimen. Although the axial strain is stable, it cannot give full play to the role of mould confining pressure. Therefore, with respect to the diameter of the test piece, we should not only ensure the role of the test mould confining pressure but also avoid the occurrence of a 'weak influence area'. It is suggested that the diameter of the specimen should be 10–20 cm.

When the height of the test piece is $h \geq 20$ cm and the diameter of the test piece is $\Phi \geq 10$ cm, the influence of the gravel particle size on the test result tends to be stable, and the size effect of the test piece has little influence on the numerical test results. At the same time, considering the simulation degree and calculation speed of the numerical test, it is suggested that the size of the specimen should be 10 cm ($\Phi$) × 20 cm ($h$).

## 3.3 Reliability verification

The reliability of the numerical simulation method was verified by taking a class A configuration in Table 2 as an example.

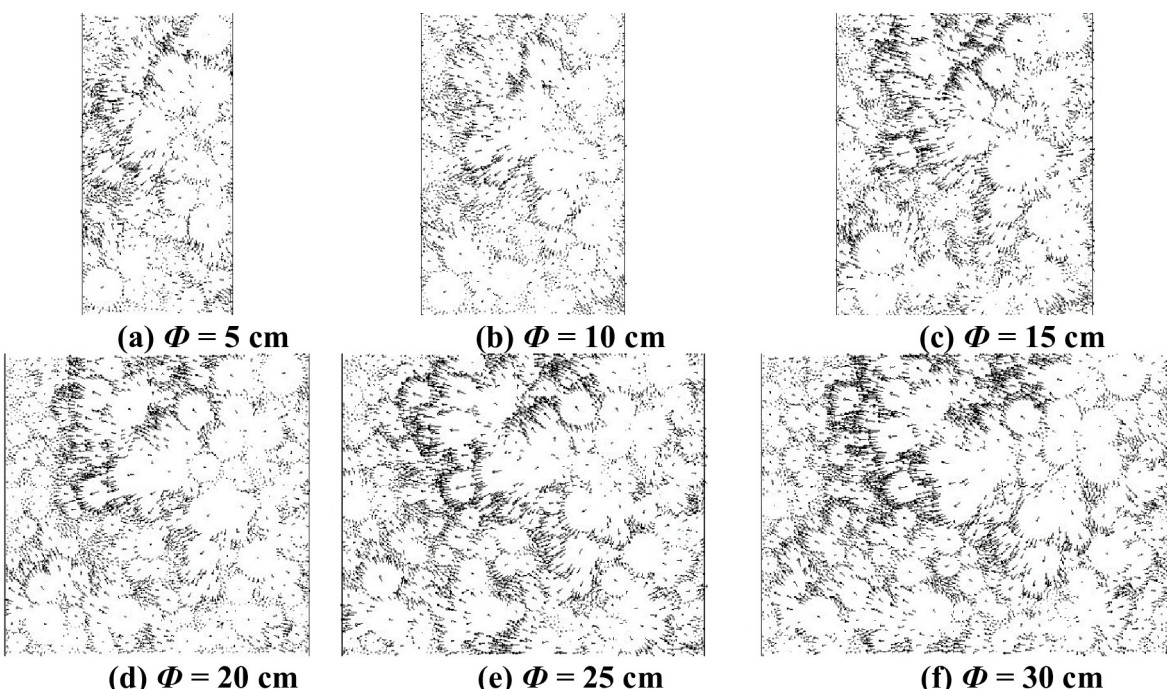

**Fig 12. Displacement vector micrograph of specimens with different diameters.** (a) $\Phi = 5$ cm. (b) $\Phi = 10$ cm. (c) $\Phi = 15$ cm. (d) $\Phi = 20$ cm. (e) $\Phi = 25$ cm. (f) $\Phi = 30$ cm.

**3.3.1 Simulation of the trial mould.** According to the research on the dynamic triaxial numerical test of graded crushed stone, the size of the test model should be 10 cm ($\Phi$) × 20 cm ($h$). The DEM command 'wall' was used to generate two vertical 'walls' with a length of 20 cm and two horizontal 'walls' with a length of 10 cm to form a closed rectangle to simulate the test mould.

**3.3.2 Generation of graded crushed stone.** The equivalent conversion method was used to convert the aggregate volume into a two-dimensional mapping area in numerical experiments. Here, the two-dimensional mapping area of 19–31.5 mm gravel is calculated as an example.

$$S_1 = \frac{DhKP_1}{\rho_1}\rho_{max} = \frac{10 \times 20 \times 0.98 \times 0.27}{2.712} \times 2.3 = 44.9 \ (\text{cm}^2).$$

The DEM command 'ball' was used to generate particles with diameters ranging from 19 to 31.5 mm. When the total area reached 44.9 cm$^2$, the particles were stopped. By analogy, the generation of graded crushed stone is completed.

**3.3.3 Reliability verification.** The indoor dynamic triaxial test results and simulation results of graded crushed stone are shown in Fig 13.

It can be seen from Fig 13 that the maximum error of the numerical and laboratory tests with a confining pressure of 50 kPa is 7%, and the average error is 3%. The maximum error in the numerical and indoor tests with a confining pressure of 100 kPa is 5%, and the average error is 2%. The maximum error in the numerical and indoor tests with a confining pressure of 150 kPa is 8%, and the average error is 1%. The results of the numerical and laboratory tests are in good agreement, which proves that the dynamic triaxial numerical test method is reliable.

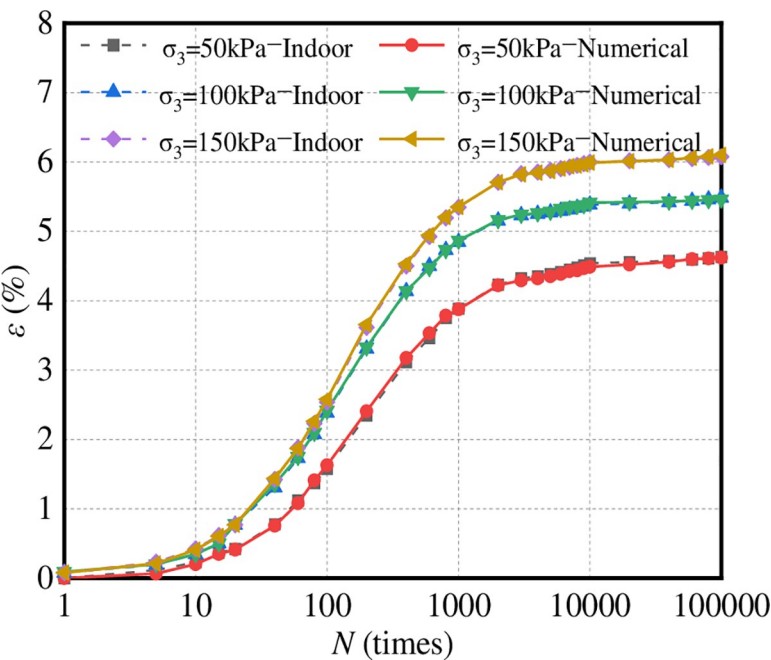

**Fig 13. Comparison between numerical simulation and laboratory tests.**

## 4 Numerical simulation of plastic deformation behaviour of graded crushed stone

In this paper, considering the permanent deformation of graded crushed stone under different stress levels and the gradual accumulation of the permanent deformation of graded crushed stone with the increase in loading times, a constitutive model of the relationship between the number of loading actions for elucidating the comprehensive action of stress level and the cumulative permanent deformation is proposed.

### 4.1 Law of plastic deformation

**4.1.1 Linear analysis of plastic deformation.** The plastic deformation curve of graded crushed stone is shown in Fig 14; the confining pressure is taken as 50 kPa.

It can be seen from Fig 14 that the relationship between the plastic deformation curve and the number of loading actions varies with the stress. The curve of the relationship between the axial strain of graded crushed stone and the number of loading actions $n$ is mainly distributed in three types of deformation regions: stable section, critical section and failure zone. The strain curves in each zone are called stable, critical and failure curves, respectively.

**4.1.2 Cumulative equation of plastic deformation of graded crushed stone.** Through the linear analysis of plastic deformation, the plastic deformation of graded gravel increases with the increase in number of loadings. The equation should meet the boundary conditions of $N = 0, \varepsilon = 0$ and $N = \infty, \varepsilon = \varepsilon_\infty$ assuming a cumulative plastic deformation equation for graded gravel. According to the boundary conditions, the cumulative equation of the plastic deformation of graded crushed stone is analysed and established.

$$\varepsilon = \varepsilon_\infty - \frac{\varepsilon_\infty}{\xi \cdot N + 1},$$

(6)

where $\varepsilon$ is the axial strain of graded crushed stone (%), $\varepsilon_\infty$ is the ultimate permanent strain of

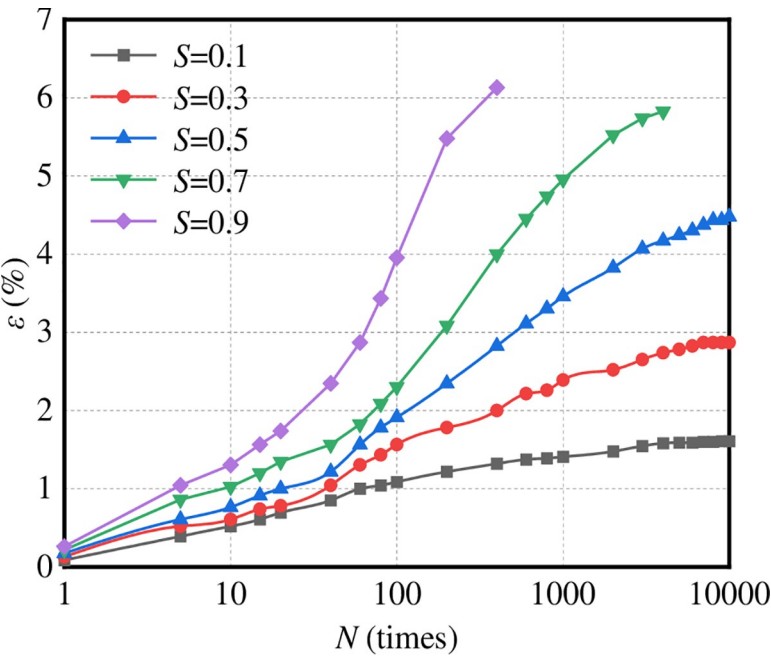

**Fig 14. Plastic deformation curves of graded crushed stone.**

graded crushed stone (%), $\xi$ is the cumulative coefficient of plastic deformation and $N$ is the dynamic load action times.

According to formula (6), the plastic deformation data of graded crushed stone are fitted, and the ultimate permanent strain and cumulative plastic deformation coefficient of graded crushed stone under three confining pressures (50, 100 and 150 kPa) are obtained. The results are given in Table 3. $R^2$ is the correlation coefficient.

## 4.2 Critical failure stress

Based on the above analysis, the upper limit of dynamic stress corresponding to the critical interval is the critical failure stress $\sigma_{\mathrm{p}}$. If the dynamic stress of the graded crushed stone base is

**Table 3. Cumulative equation of plastic deformation of graded crushed stone.**

| $\sigma_3$ (kPa) | $\sigma_d$ (kPa) | S | $\varepsilon_\infty$ (%) | $\xi$ | $R^2$ |
|---|---|---|---|---|---|
| 50 | 85 | 0.1 | 1.6520 | 0.0263 | 0.985 |
| | 256 | 0.3 | 2.8808 | 0.0112 | 0.983 |
| | 427 | 0.5 | 4.6504 | 0.0052 | 0.993 |
| | 597 | 0.7 | 6.1558 | 0.0059 | 0.997 |
| | 768 | 0.9 | 7.5340 | 0.0118 | 0.991 |
| 100 | 128 | 0.1 | 1.7398 | 0.0264 | 0.982 |
| | 384 | 0.3 | 3.4046 | 0.0080 | 0.991 |
| | 641 | 0.5 | 5.5168 | 0.0077 | 0.996 |
| | 897 | 0.7 | 7.2117 | 0.0094 | 0.998 |
| | 1153 | 0.9 | 8.8287 | 0.0150 | 0.996 |
| 150 | 170 | 0.1 | 1.9505 | 0.0113 | 0.988 |
| | 509 | 0.3 | 3.7505 | 0.0096 | 0.994 |
| | 849 | 0.5 | 6.1355 | 0.0074 | 0.991 |
| | 1189 | 0.7 | 7.6899 | 0.0163 | 0.994 |
| | 1528 | 0.9 | 9.4709 | 0.0348 | 0.998 |

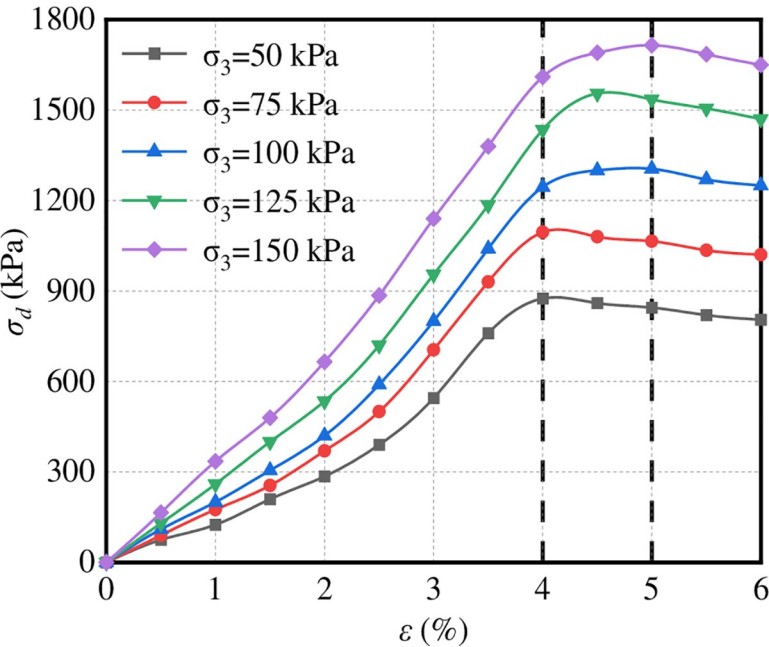

**Fig 15. Stress–strain curves of graded crushed stone.**

less than the critical failure stress, the plastic deformation of the graded crushed stone base will be effectively controlled. In this work, the plastic strain accumulation equation of graded crushed stone was fitted through the formation mechanism of plastic deformation and the dynamic triaxial test results of plastic deformation. The ultimate strain of graded gravel was regressed by SPSS software, and the critical failure stress was determined.

**4.2.1 Plastic deformation failure criteria.**   To determine the value of plastic deformation failure standard of graded crushed stone, the stress–strain relationship of graded crushed stone was studied, as shown in Fig 15.

Fig 15 shows that the stress of graded crushed stone increases with the increase in strain initially, then decreases with the increase in strain after reaching the peak value and finally tends to be stable. The failure strain corresponding to the peak value is mainly in the range of 4%–5%, which indicates that the antideformation performance of graded crushed stone decreases rapidly when the axial strain is 4%–5%. According to a previous study, an axial plastic strain of more than 4% in the pavement is unacceptable [26]. Therefore, 4% axial plastic deformation was used as the failure criterion $\varepsilon_p$ to determine the critical failure stress of graded crushed stone.

**4.2.2 Determination of critical failure stress.**   The schematic of the critical failure stress of graded crushed stone according to the limited plastic strain data of the regression graded crushed stone in Table 3 is shown in Fig 16. The failure strain line of graded broken stone is drawn in the figure, as shown by the dotted line in the figure.

It can be seen from Fig 16 that under the same confining pressure, the ultimate plastic strain increases with the increase in deviator stress, and the strain rate decreases gradually; under the same deviator stress, the ultimate plastic strain decreases with the increase in confining pressure. Confining pressure has a great influence on critical failure stress. The greater the confining pressure is, the greater the deviator stress required to reach the same strain. The critical failure stress and confining pressure are approximately linear. See Table 4 for the critical failure stress corresponding to different confining pressures.

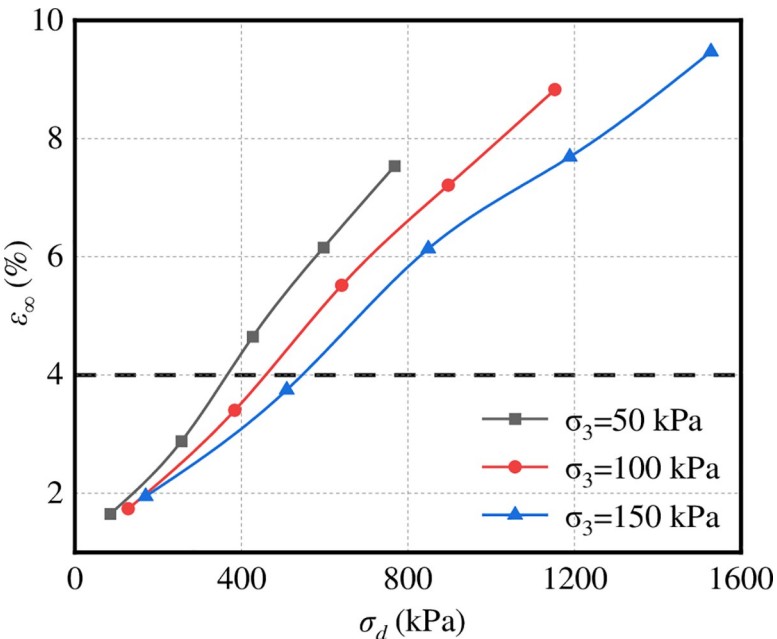

**Fig 16. Diagram of critical failure stress.**

To sum up, to effectively control the plastic deformation of graded crushed stone under repeated dynamic loading, the deviator stress under different confining pressures $\sigma_d$ should be less than the corresponding critical failure stress $\sigma_p$. The critical failure stress level corresponding to the critical failure stress $S_p$ is approximately 0.310–0.424.

## 4.3 Cumulative failure equation of plastic deformation

**4.3.1 Cumulative failure stress condition of plastic deformation.** The horizontal stress range of the graded crushed stone transition layer should be considered in the selection of confining pressure. According to ANSYS mechanical analysis, the horizontal stress range of the graded crushed stone transition layer is 48.9–70.8 kPa, and the horizontal stress range of the graded crushed stone base is 26.3–56.7 kPa. Therefore, in the dynamic triaxial simulation test of graded gravel, the range of confining pressure was taken as 20–100 kPa to cover the horizontal stress range of the graded gravel transition layer. We did not take the four levels 20, 50, 80 and 100 kPa.

It can be seen from Table 4 that to ensure failure, the cumulative failure stress level of plastic deformation used for calculation should be greater than the upper limit of critical failure stress level, i.e. $S_d > 0.424$. Test efficiency and accuracy should also be considered in the determination of $S_d$. If $S$ is too small, the number of dynamic load actions required to reach the failure strain of graded crushed stone specimens is very large, and the simulation test time is too long. If $S$ is too large, it is inconsistent with the actual stress condition of the graded gravel transition layer. In view of this, $S_d$ is proposed to be 0.5, 0.55, 0.6, 0.65 and 0.7.

**Table 4. Critical failure stress of graded crushed stone.**

| $\sigma_3$ (kPa) | $\varepsilon_p$ (%) | $\sigma_p$ (kPa) | $S_p$ |
|---|---|---|---|
| 50 | 4 | 362 | 0.424 |
| 100 | 4 | 451 | 0.352 |
| 150 | 4 | 526 | 0.310 |

**4.3.2 Determination of the cumulative failure equation of plastic deformation.** Assuming that there is no cumulative failure limit of plastic deformation for graded crushed stone materials, the cumulative failure equation of plastic deformation should satisfy the following boundary conditions:

$$\text{when } S_d = 1, N_d = 1, \tag{7}$$

$$\text{when } S_d = 0, N_d \rightarrow \infty, \tag{8}$$

where $S_d$ is the cumulative failure stress level of different plastic deformations and $N_d$ is the number of dynamic load actions required to reach the failure strain of graded crushed stone simulation specimens.

No plastic deformation cumulative failure equation strictly satisfies these two boundary conditions and can well fit all the test results. Therefore, these two boundary conditions are often relaxed to obtain a more reasonable cumulative failure equation of plastic deformation:

$$\lg S_d = \lg\alpha - \beta\lg N_d, \tag{9}$$

where $\alpha$ and $\beta$ are the undetermined regression coefficients.

Eq (9) satisfies the boundary condition given by Eq (8). See Fig 17 for $S_d$–$N_d$, $\alpha$ and $\beta$.

It can be seen from Fig 17 that the regression coefficient $\alpha$ represents the intercept of the equation curve on the ordinate axis and $\beta$ represents the slope of the equation curve. The larger the value of $\alpha$, the stronger the resistance to plastic deformation of the graded crushed stone under the action of dynamic load; the smaller the value of $\beta$, the stronger the resistance of graded crushed stone to plastic deformation.

## 5 Conclusions

1. In this work, the dynamic triaxial numerical test method of graded crushed stone based on particle flow theory and its application were studied. On the basis of DEM, a dynamic triaxial numerical test method for graded crushed stone was constructed, and the sensitivity of test conditions to numerical test results was studied. Then, the standard test conditions of the numerical test were proposed. Results show that the stability of the dynamic triaxial numerical test of graded crushed stone is better when the calculation time step is greater than 1E−4 s per step. The dynamic load has a disturbing effect on all parts of the aggregates of the virtual specimen. With increase in the number of dynamic loadings, the plastic deformation of graded crushed stone gradually accumulates, and the plastic deformation of the simulated specimen tends to be stable under 10,000 cyclic dynamic loadings. When the specimen's size is 10 cm (Φ) × 20 cm (h), the effect of its size can be ignored.

2. The dynamic triaxial numerical test method was used to study the plastic deformation law of graded crushed stone. The horizontal stress of the graded crushed stone transition layer was calculated using ANSYS. The peak deviator and critical failure stresses of graded crushed stone were determined using the numerical test method. Results show that the horizontal stress range of the graded crushed stone transition layer is 48.9–70.8 kPa, the horizontal stress range of the graded crushed stone base is 26.3–56.7 kPa, and the cumulative failure test stress level of plastic deformation should be greater than 0.424.

3. On the basis of the dynamic triaxial test method, the cumulative failure equation of the plastic deformation of graded crushed stone was established, and the coefficient of the cumulative failure equation was regressed. Results show that confining pressure has a great influence on critical failure stress. The greater the confining pressure is, the greater the

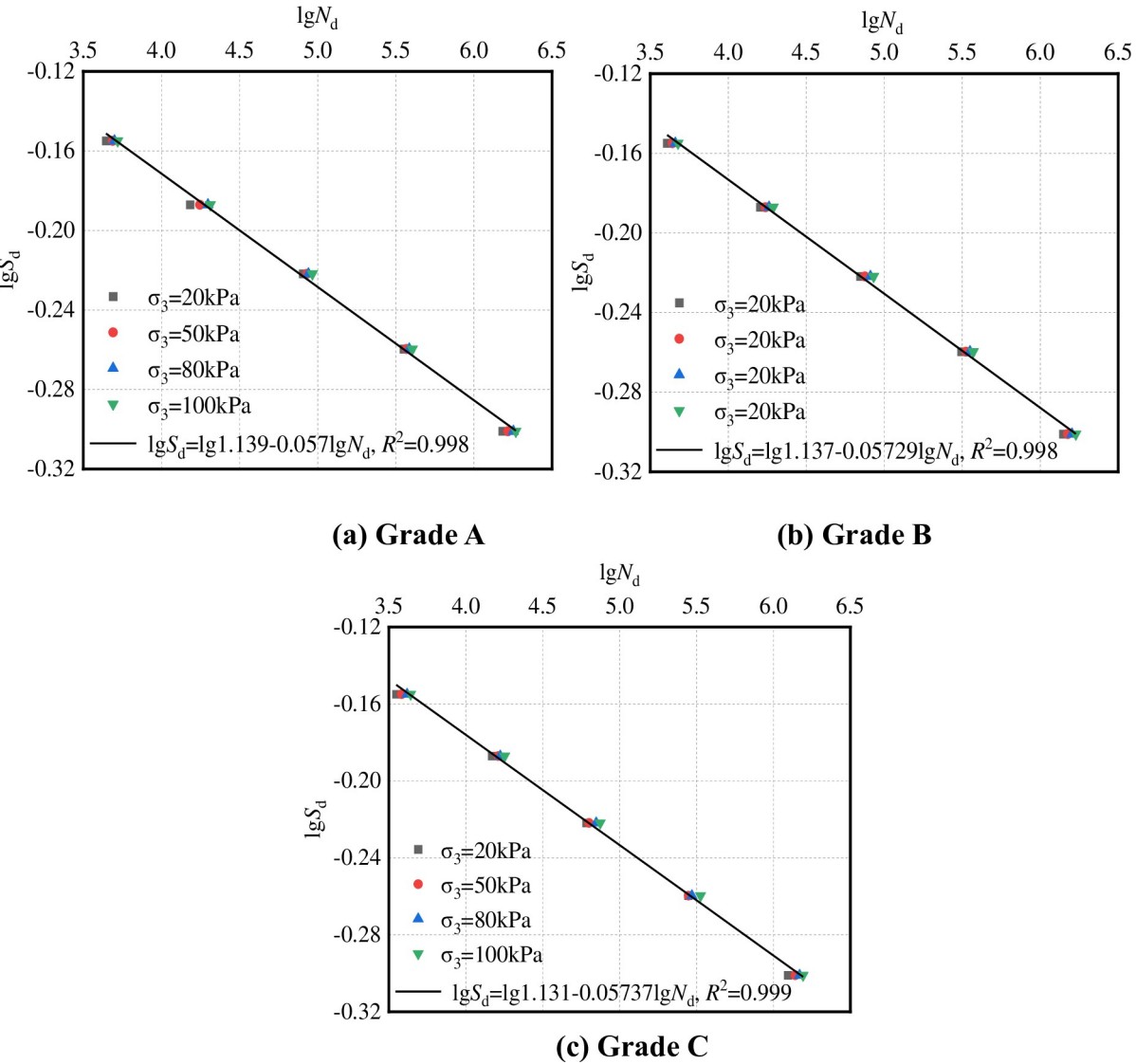

**Fig 17. Cumulative failure equation of plastic deformation.** (a) Grade A. (b) Grade B. (c) Grade C.

deviator stress required to reach the same strain. The critical failure stress and confining pressure are approximately linear, and the plastic deformation of graded crushed stone is established. The cumulative failure equation is $\lg S_d = \lg\alpha - \beta\lg N_d$.

## Author Contributions

**Conceptualization:** Yong Yi.

**Funding acquisition:** Ying-jun Jiang.

**Investigation:** Chen-yang Ni, Yu Zhang.

**Software:** Chen-yang Ni, Yong Yi.

**Writing – original draft:** Chen-yang Ni.

**Writing – review & editing:** Ying-jun Jiang, Yu Zhang.

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
