## [Decision Letter · Decision Letter 0]

23 Jun 2021

PONE-D-21-18667

Numerical Investigation of the Plastic Deformation Behaviour of Graded Crushed Stone

PLOS ONE

Dear Dr. Ni,

Thank you for submitting your manuscript to PLOS ONE. After careful consideration, we feel that it has merit but does not fully meet PLOS ONE’s publication criteria as it currently stands. Therefore, we invite you to submit a revised version of the manuscript that addresses the points raised during the review process.

Please consider all comments

We look forward to receiving your revised manuscript.

Kind regards,

Ahmed Mancy Mosa, Ph.D.

Academic Editor

PLOS ONE

Journal Requirements:

1. Please ensure that your manuscript meets PLOS ONE's style requirements, including those for file naming. The PLOS ONE style templates can be found athttps://journals.plos.org/plosone/s/file?id=wjVg/PLOSOne_formatting_sample_main_body.pdf and https://journals.plos.org/plosone/s/file?id=ba62/PLOSOne_formatting_sample_title_authors_affiliations.pdf

'Founding: This work was supported by the Scientific Research of Central

Colleges of China for Chang'an University (No. 300102218212); the scientific

project from Shaanxi Provincial Communication No. 19-27K and the scientific

project from Shaanxi Provincial Communication No. 18-02K. The authors

gratefully acknowledge this financial support.'

'No'

'I have read the journal's policy and the authors of this manuscript have the following

competing interests'

Additional Editor Comments (if provided):

Reviewers' comments:

Reviewer's Responses to Questions

**Comments to the Author**

1. Is the manuscript technically sound, and do the data support the conclusions?

Reviewer #1: Yes

Reviewer #2: Yes

2. Has the statistical analysis been performed appropriately and rigorously? 

Reviewer #1: Yes

Reviewer #2: Yes

3. Have the authors made all data underlying the findings in their manuscript fully available?

Reviewer #1: Yes

Reviewer #2: Yes

4. Is the manuscript presented in an intelligible fashion and written in standard English?

Reviewer #1: Yes

Reviewer #2: Yes

5. Review Comments to the Author

Reviewer #1: The article entitled “Numerical Investigation of the Plastic Deformation Behaviour of Graded Crushed Stone” aims to propose a numerical model of the dynamic triaxial test of graded crushed stone to study its dynamic characteristics.

The paper is well written and properly organized. The English is correct.

The authors introduce the paper correctly and explain the study. However, I also have some major and minor queries:

Major queries:

1. The abstract is not clear. It is complicated to understand what statements are results and conclusions of this research. Also, the method is not clear. It should be re-written to properly describe the problem, the method and the results/conclusions. Formulae should not be included.

2. In the introduction, the authors cite several papers but they do not describe the results or conclusions of these research works, which could be useful to define the objectives of this paper.

3. It is complicated to perform triaxial test on stone crushed samples. The suitability of the method is not very clear. A photo of the samples could help to understand and identify the soil considered. Moreover, the references, properly described, could support the method and objectives of this paper.

Minor queries:

4. The DEM significance is not described in the manuscript, only in the abstract.

My recommendation is major revision.

Reviewer #2: General comment

The manuscript focuses on dynamic characteristics of graded crushed stone. It is interesting to see a numerical study of dynamic triaxial test to study the accumulation rule and failure standard of permanent deformation. Good expression and rigorous data analysis have been made in this paper, while there are some detials need to be modefied. I have indicated those defects as specific comments. I hope the authors will spend some time to deal with those detials. With this and other concerns. I recommend to accept the paper for publication.

Specific comments:

1.The abstract should be rewritten to explain the importance and urgency of your research firstly, then summarize the evaluation result at the multiple temporal and spatial scale.

2.The authors indicated that Confining pressure was determined by ANSYS, so the finite element modeling and calculation need to be illustrated.

6. PLOS authors have the option to publish the peer review history of their article (what does this mean?). If published, this will include your full peer review and any attached files.

Reviewer #1: No

Reviewer #2: No

---

## [Author Response · Author response to Decision Letter 0]

21 Aug 2021

Response to Reviewers

Dear editor:

On behalf of my co-author, I would like to thank the editorial department and the reviewers for their positive and constructive opinions and suggestions on our topic " Numerical Investigation of the Plastic Deformation Behaviour of Graded Crushed Stone" (PONE-D-21-18667). We have carefully studied these suggestions and made some amendments to the paper. We have tried our best to revise the manuscript as required. After carefully considering the opinions of the editor and the reviewers, the main corrections and answers to the reviewers are as follows:

Reviewer #1: The article entitled “Numerical Investigation of the Plastic Deformation Behaviour of Graded Crushed Stone” aims to propose a numerical model of the dynamic triaxial test of graded crushed stone to study its dynamic characteristics.

The paper is well written and properly organized. The English is correct.

The authors introduce the paper correctly and explain the study. However, I also have some major and minor queries:

Major queries:

1. The abstract is not clear. It is complicated to understand what statements are results and conclusions of this research. Also, the method is not clear. It should be re-written to properly describe the problem, the method and the results/conclusions. Formulae should not be included.

The abstract has been re-written.

2. In the introduction, the authors cite several papers but they do not describe the results or conclusions of these research works, which could be useful to define the objectives of this paper.

The following abstract in the introduction of this paper is just to prove the general applicability of DEM method, and its purpose has been added

3. It is complicated to perform triaxial test on stone crushed samples. The suitability of the method is not very clear. A photo of the samples could help to understand and identify the soil considered. Moreover, the references, properly described, could support the method and objectives of this paper.

Thank the reviewers for their suggestions.

Minor queries:

4. The DEM significance is not described in the manuscript, only in the abstract.

My recommendation is major revision.

DEM has been widely used in scientific research, the author thinks that it is not necessary to spend a lot of paper to write.

Reviewer #2: General comment

The manuscript focuses on dynamic characteristics of graded crushed stone. It is interesting to see a numerical study of dynamic triaxial test to study the accumulation rule and failure standard of permanent deformation. Good expression and rigorous data analysis have been made in this paper, while there are some detials need to be modefied. I have indicated those defects as specific comments. I hope the authors will spend some time to deal with those detials. With this and other concerns. I recommend to accept the paper for publication.

Specific comments:

1.The abstract should be rewritten to explain the importance and urgency of your research firstly, then summarize the evaluation result at the multiple temporal and spatial scale.

Modified.

2.The authors indicated that Confining pressure was determined by ANSYS, so the finite element modeling and calculation need to be illustrated.

Combined with the purpose of the paper and save space, ANSYS software did not start to explain.

Sincerely Yours,

Chenyang Ni

---

## [Decision Letter · Decision Letter 1]

20 Sep 2021

Numerical Investigation of the Plastic Deformation Behaviour of Graded Crushed Stone

PONE-D-21-18667R1

Dear Dr. Ni,

We’re pleased to inform you that your manuscript has been judged scientifically suitable for publication and will be formally accepted for publication once it meets all outstanding technical requirements.

Kind regards,

Ahmed Mancy Mosa, Ph.D.

Academic Editor

PLOS ONE

Additional Editor Comments (optional):

Reviewers' comments:

Reviewer's Responses to Questions

**Comments to the Author**

1. If the authors have adequately addressed your comments raised in a previous round of review and you feel that this manuscript is now acceptable for publication, you may indicate that here to bypass the “Comments to the Author” section, enter your conflict of interest statement in the “Confidential to Editor” section, and submit your "Accept" recommendation.

Reviewer #1: All comments have been addressed

2. Is the manuscript technically sound, and do the data support the conclusions?

Reviewer #1: Yes

3. Has the statistical analysis been performed appropriately and rigorously? 

Reviewer #1: Yes

4. Have the authors made all data underlying the findings in their manuscript fully available?

Reviewer #1: Yes

5. Is the manuscript presented in an intelligible fashion and written in standard English?

Reviewer #1: Yes

6. Review Comments to the Author

Reviewer #1: The authors have addressed all the coments proposed. There are no furher comments. My recommendation is accept.

7. PLOS authors have the option to publish the peer review history of their article (what does this mean?). If published, this will include your full peer review and any attached files.

Reviewer #1: No

---

## [Editor Report · Acceptance letter]

23 Sep 2021

PONE-D-21-18667R1 

Numerical Investigation of the Plastic Deformation Behaviour of Graded Crushed Stone 

Dear Dr. Ni:

I'm pleased to inform you that your manuscript has been deemed suitable for publication in PLOS ONE. Congratulations! Your manuscript is now with our production department. 

Kind regards, 

on behalf of

Dr. Ahmed Mancy Mosa 

Academic Editor

PLOS ONE